# RD²BENCH: TOWARDS DATA-CENTRIC AUTOMATIC R&D

## ABSTRACT

The progress of humanity is driven by those successful discoveries accompanied by countless failed experiments. Researchers often seek potential solutions described in related literature (raw information) and verify them through experiments. With the explosive growth of deep learning literature and methods, such a process imposes a more significant burden on researchers and renders successful discoveries veiled. Therefore, automating such a research and development (R&D) process is an urgent need. In this paper, we serve as the first effort to formalize the goal by proposing a **R**eal-world **D**ata-centric automatic **R&D Bench**mark, namely RD²Bench. RD²Bench benchmarks the whole data-centric automatic R&D (D-CARD) process, including extracting methods (formulas and models) from raw information (reports and papers) and implementing methods through codes. Specifically, to investigate the capability boundaries of the state-of-the-art (SOTA) large language models (LLMs) in the unexplored D-CARD, we conduct exhausting and expensive human annotations and experiments. We evaluate the performance of SOTA LLMs on our identified 27 formulas and 6 models across various difficulty levels from financial reports and ML papers. We find that although RD²Bench is very challenging, SOTA LLMs possess promising potential to bring more significant development to D-CARD. We appeal to research teams with various domain expertise to consider constructing domain-specific D-CARD benchmarks, contributing to both a cross-domain D-CARD platform and the potential revolutionary upgrade to human productivity.

## 1 INTRODUCTION

*"I have not failed. I've just found 10,000 ways that won't work."*

*— Thomas Alva Edison*

The advancement of human society and the enhancement of living standards are highly correlated with the development of technology (Smith, 1937; Ranis & Fei, 1961; Perez, 2003; Brynjolfsson & McAfee, 2014). Numerous truths and principles remain undiscovered in the world, awaiting experimental exploration (Shapere, 1964; Popper, 2005). Those few successful discoveries, accompanied by countless failed experiments, propel the frontiers of technology. Historically, scientific researchers, including Edison, have undertaken extensive experiments by conducting them manually. In the age of AI, the influence of data-driven solutions, such as machine learning (ML) systems, is rapidly expanding (Mikolov et al., 2013; Devlin et al., 2018; OpenAI, 2023b). These systems are known for their robust fitting capabilities and their "black box" nature, which significantly increases the experimental load on researchers and hinders the process of identifying and validating effective methodologies. This paper concentrates on this critical scenario, which we refer to as *Data-Centric Research and Development (R&D)*. To cope with the prohibitively expensive costs and the overwhelming volume of experiments required, we consider automating such an R&D process for higher research efficiency by leveraging the strong language understanding and programming ability of the state-of-the-art (SOTA) large language models (LLMs) (Srivastava et al., 2023). The brief illustration of **D**ata-**C**entric **A**utomatic **R&D** (D-CARD) is shown in Figure 1.

The first step towards automatic R&D is to formalize the task and provide a benchmark for identifying the potential effective methods and research directions. Intuitively, an outstanding methodology

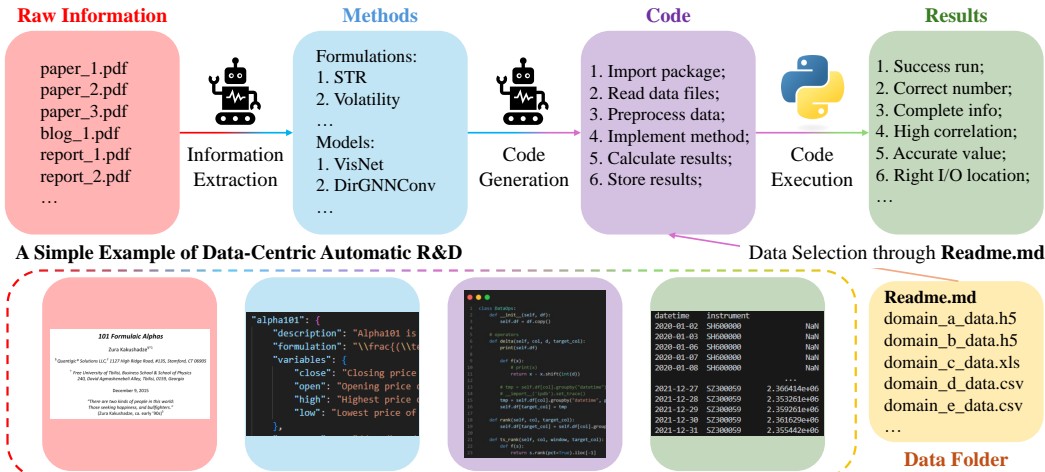

Figure 1: An overview of the R&D process. Researchers read papers and reports to extract the implementable methods (usually formulated as mathematical formulas or model architectures) for seeking potential research directions. Then, they accurately implement the methods to obtain the results for further analysis and development.

identified by the benchmark should possess (1) strong **language understanding ability** to identify the implementable methods or ideas (e.g., formulations and models) in the given raw information (e.g., papers, reports, websites, etc.) and (2) strong **implementation ability** to accurately implement the methods by programming and then obtain reliable experimental results. Previous work focuses on benchmarking the different aspects of the two abilities. Specifically, the language understanding ability of LLMs is partly evaluated through analyzing their performance on relation extraction (Wadhwa et al., 2023), question answering (Zhuang et al., 2023), and other natural language processing (NLP) tasks (Qin et al., 2023a). Meanwhile, the implementation ability of LLMs is partly tested through benchmarks like SWE-Bench (Jimenez et al., 2023b), ToolBench (Qin et al., 2023c), ML-Bench (Liu et al., 2023b) and MetaTool (Huang et al., 2024), which study their ability of solving GitHub issues, using tools to program, and determining whether to use tools in a given scenario.

In this paper, we serve as the first effort to investigate the capabilities of the SOTA LLMs in tackling automatic R&D and propose a **R**eal-world **D**ata-centric automatic **R&D Bench**mark ($RD^2Bench$). The scenario studied by $RD^2Bench$ possesses two unique and distinct characteristics that fundamentally differentiate it from others. First, $RD^2Bench$ focuses on studying the real-world scenario where all the operations in R&D are automatic and evaluated as a whole, thus navigating the related future research efforts toward the goal of developing human technology more effectively. The real-world scenario requires more comprehensive and advanced model capabilities and exhibits new challenges. Second, we study the real-world automatic R&D in data-centric settings to navigate future work toward the urgent experimental exploration need brought by black-box data-driven models. Compared with existing benchmarks, $RD^2Bench$ possesses two significant advantages:

(1) **$RD^2Bench$ evaluates the interaction and synergistic effects of various model capabilities** instead of focusing on a single aspect of ability, which not only captures the frontier of SOTA LLMs but also bridges the gap between studying "individual ability" and "real-world synergistic effects of abilities". In automatic R&D, an ML system fails to complete the task even if it possesses both the strong information extraction ability and the strong programming or tool-using ability: While it succeeds in extracting methods and implementing them, it fails in selecting the appropriate data from the datasets or misunderstanding either the descriptions of data features or the requirements expressed by prompts. Additionally, exhaustively enumerating all the aspects for benchmarking is extremely challenging, which is overcome by $RD^2Bench$.

(2) **$RD^2Bench$ tends to select well-performing trustworthy models** instead of those models that fail to learn rationales and causality yet possess outstanding performance. Specifically, ML systems easily achieve SOTA performance on previous benchmarks by shortcut learning or learning spurious correlations instead of learning rationales or causality (Mudrakarta et al., 2018; Geirhos et al.,

2020; Cui & Athey, 2022; Wang et al., 2022; Chen et al., 2023). This renders a benchmark ineffective and misleading as it fails to accurately identify the well-performing trustworthy methods. For example, an ML system achieves SOTA performance on dog classification by merely recognizing grass (Zhang et al., 2021). RD$^2$Bench, on the contrary, eliminates such models by its high difficulty and large scope. The decision rules of models have to simultaneously satisfy at least four major requirements: (1) accurately and comprehensively extracting the implementable methods; (2) precisely selecting the method-specific data for computation; (3) correctly writing the code according to the logic expressed by methods and prompts; (4) successfully storing the correct results in a predefined format. Therefore, the decision rules of models selected by this benchmark are stable (work well in various situations), and thus getting closer to rationales and causality (Cui & Athey, 2022).

We evaluate the existing SOTA LLMs on RD$^2$Bench to expose their bottleneck and characterize the future research direction. RD$^2$Bench reveals new insights: (1) Among the popular LLMs, GPT-4 exhibits promising potency in dealing with the D-CARD task; (2) Detailed information of data descriptions significantly improves the performance of GPT-4; (3) The ability to query domain-specific knowledge is a basic requirement of D-CARD methods; (4) The more complex the method is, the more unstable the model's performance is.

## 2  RELATED WORK

### 2.1  LLM AS AUTONOMOUS AGENT

In the past few years, LLM has made great achievements in both academia and industry (OpenAI, 2023a; Touvron et al., 2023), and has achieved results that surpass the previous level in a number of classic tasks (Zhao et al., 2023). Research has shown that with the growth of data volume and model size (Zoph et al., 2022), LLM has emerged with stronger reasoning and other capabilities (Ouyang et al., 2022). These capabilities enable LLM to exhibit certain agent-like behaviors in some tasks such as using or creating tools (Qin et al., 2023b; Qian et al., 2023), planning (Yao et al., 2023; Brown et al., 2020a), and memory. Therefore, more and more researchers have expressed their expectations for its human-like and overall capabilities, and have made preliminary explorations of it as an independent agent (Wang et al., 2023a; Shinn et al., 2023b). Multi-agent collaboration (Wu et al., 2023; Li et al., 2023) is also introduced to LLM for better accuracy and generalizability. Moreover, for reducing human efforts and automatically exploring, previous work focuses on autonomous LLM agents for general purpose are purposed (Yang et al., 2023b; Shen et al., 2023). Positive views further believe that the realization of AGI may come from the evolution of autonomous LLM and some inspiring examples have been released (Penov et al., 2024).

However, most research still focuses on limited scenarios that are given with clear and fixed questions and backgrounds. A recent work (Yang et al., 2023d) has attempted to introduce LLM to the R&D field and formalize the R&D process as a sequence of tasks. However, there is no easy-to-use benchmark for the community and current R&D tasks may be too general and can't reveal significant signals. In this work, we propose a benchmark for LLM in data-centric R&D tasks and provide a comprehensive evaluation.

### 2.2  SEMI-AUTOMATIC R&D WITH AGENTS

Scientific research and development (R&D) is a time-consuming and important process. In the past, R&D has been mainly conducted by human researchers with countless failed experimental explorations and creative observation conclusions. Agents have been introduced to R&D to reduce human efforts and automatically explore. Recently, there have been attempts to partly automate R&D, including the automatic chemical synthesis planning (Boiko et al., 2023), automatic molecular design (Joshi & Kumar, 2021; Schneider, 2017; Boiko et al., 2023), automatic theorem proving (Wang et al., 2023b; Yang et al., 2023c). However, these attempts mainly focus on automatic searching for possible solutions and optimizations with symbolic representation (Lu et al., 2023) and heuristic techniques (Whalen, 2016), but less addressing long-horizon planning, implementation, and reasoning for the next step idea exploration. Moreover, the data-centric R&D tasks currently have not been explored in the community, and no benchmark has been proposed for the community. Previous works have applied LLM to real-world R&D tasks such as debugging issues (Tian et al., 2024; Jimenez et al., 2023a) or only focus on data-centric but not real-world R&D tasks (Liu et al.,

2023a). In this work, we propose a benchmark for LLMs in data-centric R&D tasks and evaluate the performance of LLMs.

## 3 RD²BENCH

Overall, our benchmark focuses on evaluating the finally implemented results according to the given raw information (e.g., papers, reports, websites, etc.). Moreover, we also provide human-annotated ground-truth information corresponding to the intermediate steps for debugging and more comprehensive evaluation. RD²Bench selects well-performing models that follow human operations and accurately calculate the final results. We introduce the details of our proposed RD²Bench in the following sections. In section 3.1 and section 3.2, we introduce how we collect data and perform human annotation to form RD²Bench. Then, we elaborate on the two necessary steps, namely method extraction and method implementation, to perform R&D in section 3.3 and section 3.4. Finally, we detail our adopted evaluation metrics in section 3.5.

As an initial step toward data-centric automatic R&D, our study focuses on the financial domain as a starting point. Our motivations are as follows: (1) **The financial domain is representative**. It heavily relies on data and has high scalability to be extended to academic research with minimal adjustments. In the future, we plan to expand the reports in the current dataset to include research papers (e.g., papers scraped from OpenReview). The methods will include models and formulas from papers, and our current manually implemented code could be replaced by GitHub code from open-source papers. At that point, we could benchmark a model's capability to conduct ML research. (2) **The financial domain is well-defined**. We can establish well-defined academic questions in this scenario, with clear evaluation metrics and an analytical, streamlined process. The F1 score and accuracy for method extraction and implementation are core metrics indicating the development of data-driven automatic R&D. The whole process is fully traceable, making it easy to explain how each final result is achieved.

### 3.1 DATA COLLECTION

We consider the raw information that contains formulas and models, which represent a wide range of methods proposed in the AI domain.

**Data Collection with Formulas.** We prepare raw information that contains formulas as the input of R&D. Raw information is presented as publicly available financial reports and stock trading data. Formulas are usually mathematical formulas that take complex numeric input data about stock, company, and market as input and output a series of values with the time series. We collect financial reports with 27 implementable formulas distributed in three difficulty levels: easy, medium, and hard. Domain experts manually label the difficulty levels according to the complexity of implementation. To obtain their implementation results, an agent is expected to accurately select the features from three types of trading data scattered across 2010 to 2022, namely fundamental, price-volume, and high-frequency data. We denote the three types of data as Data I, II, and III, respectively.

**Data Collection with Models.** We collect papers with six open-sourced models (Gravina et al., 2023; Rossi et al., 2023; Rampášek et al., 2022; Lim et al., 2021; Yang et al., 2023a; Wang et al., 2024). The implementation of models adopts Pytorch (Paszke et al., 2019) and torch_gemometric framework (Fey & Lenssen, 2019) to perform deep learning. All the papers and models are publicly available. We manually label the difficulty level (easy, medium, hard) of the task based on the complexity of implementation (computational graphs and tensor operations). We refer the readers to the appendix for more details about the dataset and the task.

### 3.2 HUMAN ANNOTATION

To provide a more comprehensive evaluation for debugging and analyzing, we conduct human annotation to provide the ground-truth results of our collected data, namely method extraction results and method implementation results.

**Challenges.** We confront five main challenges in the human annotation process. First, we need to identify the difficulty levels of methods to ensure the diversity of our benchmark and expose the bottleneck of current models. Second, we have to identify and discard the raw information if

its presented methods demand unavailable data: The computation of some formulas can require confidential information that is not publicly available. Third, since the definitions or descriptions of some methods can be vague, leading them to be unimplementable, we have to filter out these methods. Fourth, some domain-specific methods containing factual errors should be filtered out since they are not implementable. Fifth, we should distinguish the domains and types of the methods according to their descriptions. To sum up, all the challenges imply the fact that human annotation of RD$^2$Bench requires expensive time cost and the expertise of annotators. Therefore, we commit more effort to designing the annotation guidelines, process, and quality control to ensure the dataset quality.

**Annotation Guidelines.** The annotation guidelines are discussed and formulated after the trial phase, where each annotator completes 3-5 trial annotations. Our goal is to *identify* the described methods (formulas and models) in publicly licensed raw information and then *implement* them. In the *method identification (extraction)* process, a method is identified if: (1) all required data features for its computation are present in our predefined dataset, and (2) all necessary information for reproducing its code is explicitly available in the report. For example, negative examples include instances where variables used in the formula are not declared in the report; descriptions are vague or lack critical information, making reproduction infeasible; required data features are too rare or costly to obtain, thus lacking general applicability. Methods are extracted following a predefined schema (details are described in Appendix D), and corresponding code implementations are created to reproduce the results. To enhance reproducibility, generalization, and verification, we also define the scope of data features: In a specific domain, most data features are publicly accessible while a small portion may be costly and difficult to obtain. In the *code implementation* process, if the original report provides source code that can be successfully executed, it is executed by annotators and marked as the successfully implemented code for the method. If source code is absent or not executable, annotators write and execute code for reproduction. Manually written code must follow these specifications: (1) Use Python in Jupyter Notebook; (2) Import all required packages at the beginning of the notebook; (3) Include the method information in the form of the predefined schema as comments at the start; (4) Begin with a data loading step; (5) Treat each block of code involving input data transformation or format changes as a separate step; (6) End with an output data storage step, saving results in folders named after the method; (7) Display the intermediate results of each step in the notebook; (8) Store completed notebooks in the "check" folder.

**Annotation Process.** We build an annotation team where members possess varying levels of expertise in both ML and financial domains, categorized as follows: undergraduate students, master students, doctoral students, postdoctoral researchers/general researchers in the Financial AI industry, and senior researchers in the Financial AI industry. A 2-3 hour offline training session was held, including live demonstrations and interaction, with a recorded session provided for later reference. The session covered the guidelines and a full annotation workflow demonstration by a senior researcher. After the training stage, each annotator completes 3-5 trial annotations. The trial serves two purposes: (1) refining the schema for method reproduction and (2) evaluating the quality and expertise of annotators. Trial results are rigorously reviewed by senior researchers and authors. The obtained results were applied to real financial scenarios (e.g., backtesting) for validation. The annotators who pass the trial will compose our annotation team. Special cases (e.g., whether a method can be included if parameters are inferred by annotators) are discussed and decided collectively by the team. FAQs are documented and shared with all members. More details are presented in Appendix E.

**Annotation Quality.** To ensure clarity of guidelines, as well as accuracy in results, multiple annotators annotate and implement methods for the same report. Discrepancies are analyzed to verify the robustness of the guidelines and the reliability of the results. Both the extraction and implementation results in our "ground_truth" folder show their consistency across the multiple annotations, which demonstrates the quality of both our annotation guideline and annotation results. During the annotation process, the tasks of annotators vary according to their annotating status. Specifically, method extraction and implementation tasks are assigned by senior researchers based on profiles of annotators and their trial performance. For annotators exceeding the expected annotation time, task complexity is adjusted (becomes easier) accordingly.

### 3.3 METHOD EXTRACTION STEP

We evaluate the ability of models to recognize and extract information from raw information (e.g., R&D context). A qualified model is expected to discern feasible methods (formulas and models) from extensive research materials and extract all necessary information for implementing these formulas. The ability serves as the foundational premise for subsequent code implementation.

We expect the model to accurately and comprehensively extract the methods mentioned in the research materials it reviews, including all essential conditions required to implement the method. For methods with incomplete information, further implementation is not required; for methods with complete conditions, a model is expected to correctly comprehend the semantic meaning of these conditions stated in natural language and generate corresponding code. Specifically, we have predefined the extraction format (key-value pair) for the model. We employ the F1 score to measure the comprehensiveness and accuracy of method identification and extraction.

Note that some methods in the original materials might only imply their function, effect, or origin through their names without explicitly presenting their formulas, definitions, or other details. In such cases, the model may choose not to extract them or opt to autonomously complete them based on the semantics of the original materials. We expect the latter approach in future work, as it showcases the creativity of models by proposing new formulas and generating brand-new, informative, and reliable information. In the current version of the benchmark, only methods mentioned by name are evaluated in this manner; future iterations will explore and assess the model's ability to generate new names and formulas when none are explicitly mentioned.

### 3.4 METHOD IMPLEMENTATION STEP

In this section, we evaluate the performance of LLM in the implementation of methods. Given all the necessary conditions provided to the model after the previous step, the model needs to select the necessary data and write code from scratch to implement the method with an informative and well-organized prompt. Details of the prompt are included in the dataset, which is also shown in the appendix. We encourage models to use Python and perform data analysis. They are also permitted to use common machine-learning libraries. One example of the method implementation step is shown in Figure 2.

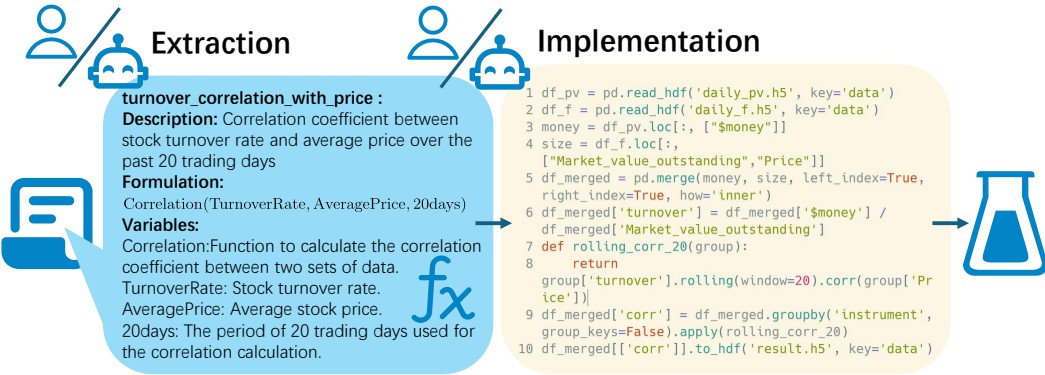

Figure 2: An example of a formula implementation task.

### 3.5 EVALUATION METRICS

We adopt multiple metrics to evaluate model performance in each step. For formula implementation, we adopt the average and maxima "running success rate", "format success rate", "Pearson correlation" and "value accuracy" across multiple independent attempts. We use "avg.", "exe.", "form.", "corr.", and "acc." to denote the average value, number of successful execution times, number of matched result formats, the correlation, and the accuracy of corresponding values, respectively. We refer the readers to more details about the metrics calculation details in App A.

For model implementation, we believe a successful implementation of a model should be consistent with the ground truth implementation as the model can be viewed as a numeric function and combination of tensor transformations. Therefore, we propose these two metrics for the model architecture implementation task: tensor shape consistency rate (tsc.), tensor value consistency rate (tvc.). Specifically, for each model layer, we calculate the consistency rate of the tensor shape and tensor value between the ground truth implementation and the implementation generated by the LLM. All the ground truth tensor values are determined by ground truth implementation codes with random Gaussian noise. Therefore, the formula for the two metrics is as follows, where $S_{\text{shape}}^i$ and $S_{\text{value}}^i$ are the consistency rate of tensor shape and tensor value in layer $i$, respectively, and $d_i$ is the maximum length of the two tensors as the two tensors are $\mathbf{Z}_i$ and $\mathbf{Z}_i^*$, the ground truth and the generated tensor, respectively:

$$S_{\text{shape}}^i(\mathbf{Z}_i, \mathbf{Z}_i^*) = \left(1 + \exp\left(\frac{\sum_{j=1}^d |\dim(\mathbf{Z}_i)_j - \dim(\mathbf{Z}_i^*)_j|}{d}\right)\right)^{-1},$$

$$S_{\text{value}}^i(\mathbf{Z}_i, \mathbf{Z}_i^*) = \left(1 + \exp\left(\frac{\sum_{j=1}^d |\mathbf{Z}_i^{(j)} - \mathbf{Z}_i^{*(j)}|}{d}\right)\right)^{-1}, d = \max(\text{len}(\dim(\mathbf{Z}_i)), \text{len}(\dim(\mathbf{Z}_i^*))),$$

(1)

while the shorter tensor is padded with zeros to match the length of the longer tensor. As the final score of the two metrics, we use the weighted sum of the consistency rate of all layers, weight increases with the depth of the layer and is summed as one: $S_{\text{final}} = \frac{\sum_{i=1}^n S^i \cdot \gamma^i}{\sum_{i=1}^n \gamma^i}$, where $n$ is the number of layers in the model, $\gamma$ is a tunable hyperparameter to control the weight increase, and we set $\gamma = 1.1$ in our experiments.

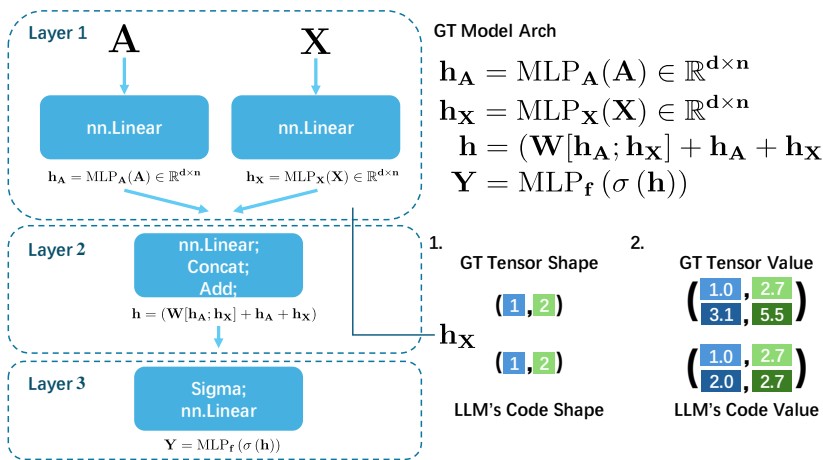

Figure 3: An example of metrics calculation for model architecture implementation task.

An example of the calculation is shown in Figure 3, using model LinkX (Lim et al., 2021) as an example. Meanwhile, we also include the "average running success rate" as the basic metric for the model architecture implementation task, which is the same as the formula implementation task.

## 4 EXPERIMENTS

### 4.1 EXPERIMENTAL SETTINGS

As we have numeric input and output in R&D tasks, we set numeric equability with 1e-6 as tolerance for the evaluation of the implementation of methods. We set the base models as GPT-4-turbo (OpenAI, 2023a), GPT-4-32k (OpenAI, 2023a), GPT-35-turbo-16k (OpenAI, 2023a) and Llama2 (Touvron et al., 2023) for the experiments. All the methods mentioned above, and their corresponding results are executed with Azure OpenAI API. There is no external data, resources, human feedback, or internet access involved in the experiments. We perform 20 independent attempts for each model and calculate the average and maximum value of each metric. As most of our input data is encoded in the form of document files, we first use parsing tools to extract text content from files. Azure document intelligence API (4.0) is used for parsing reports and academic papers in PDF format.

## 4.2 RESULTS OF METHOD EXTRACTION

We evaluate the information extraction ability of models. As shown in Table 1, we observe that GPT-4-turbo, GPT-4-32k, LLaMa3-70b, and LLaMa2-70b achieve competitive performance, which makes them possible to perform information extraction automatically. The performance of the four foundation models is stronger than of Phi3-4k, indicating more future endeavors to improve the extraction ability of Phi3-4k.

## 4.3 RESULTS OF FORMULA IMPLEMENTATION

In this section, we compare the performance of different models in the model architecture implementation task. We use the proposed metrics to evaluate the performance of the models. The results are shown in Table 2 and Table 3. We observe that the GPT-4-turbo achieves better performance than GPT-35-turbo and Phi3-128k in the model architecture implementation task. Overall experimental results indicate ample room for further research on the difficulty of the task and the challenges in automating R&D tasks. Specifically, we obtain the following four major findings revealed by the experimental results.

| Metrics | Precion | Recall | F1 |
|---|---|---|---|
| GPT-4-turbo | 0.818 | 0.800 | 0.809 |
| GPT-4-32k | 0.818 | 0.818 | 0.818 |
| LLaMa3-70b | **0.909** | 0.833 | **0.869** |
| LLaMa2-70b | 0.818 | **0.900** | 0.857 |
| Phi3-4k | 0.636 | 0.750 | 0.688 |

Table 1: Results of method extraction.

| Agentic Workflows | Avg. Exec. | Avg. Format | Avg. Corr. | Max. Corr. |
|---|---|---|---|---|
| Few-shot Brown et al. (2020b) | 0.733 | **0.433** | 0.454 | 0.562 |
| CoT Wei et al. (2022) | **0.833** | **0.433** | 0.336 | 0.538 |
| Reflexion Shinn et al. (2023a) | 0.822 | 0.400 | 0.269 | 0.550 |
| Self-Debugging Chen et al. (2024) | 0.367 | 0.256 | 0.232 | 0.539 |
| Self-Planning Jiang et al. (2023) | 0.578 | 0.211 | 0.119 | 0.341 |
| GPT-4-turbo | 0.798 | 0.378 | **0.568** | **0.835** |

Table 2: The performance of agentic workflows on RD$^2$Bench. All the agentic workflows are based on GPT-4-turbo due to its overall best performance across RD$^2$Bench.

**LLM agents hold promising potential to tackle D-CARD.** We can observe from Table 2 and Table 3 that GPT-4 possesses the ability to tackle some simple D-CARD cases without adopting any additional techniques. Specifically, GPT-4 achieves a high maximum correlation coefficient with the ground-truth results in implementing both easy and medium formulations: GPT-4-turbo achieves the maximum correlation value in implementing easy and medium formulas. However, GPT-4 fails to precisely match the exact ground-truth values due to some minor mistakes, such as missing the domain common knowledge (e.g., using percent change rather than difference when calculating growth), mismatching the output format, and unnecessarily introducing additional computational operations.

**Precisely understanding and selecting data requires more detailed data information in D-CARD.** As shown in Table 6, we observe a special situation where GPT-4 significantly fails to implement a simple formulation while succeeding in implementing the harder ones. After analyzing its generated code, we find that GPT-4 confuses the different semantic meanings of data features due to their close natural language descriptions, which renders the subsequent calculation ineffective. For example, GPT-4 confuses the two terms named "volume" and "volatility" and always opts to use "volume" data when "volatility" is required. If we manually improve our initial prompt by adding a more detailed description, GPT-4 succeeds in understanding the semantic difference and obtains over 99% performance in the accuracy of values.

**The ability to query domain-specific knowledge is a basic requirement of D-CARD methods.** As we mentioned in the first finding, missing domain common knowledge impedes GPT-4 from calculating precisely matched final results. Additionally, we find that the implementation of some operations in a formulation also requires domain-specific knowledge. For example, in the financial domain, it's clear enough for financial practitioners to implement the operation named "IndNeutralize(x,g)" by merely giving the description "x cross-sectionally neutralized against groups g". However, in the code generated by GPT-4, it defines a function named "IndNeutralize(series, indus-

try)" and leaves its content blank by merely adding a notation "Please replace this with your actual function definition".

**The more complex the method is, the more unstable the model performance is.** As shown in the columns of Table 6 named "avg. exec.", "avg. form.", and "avg. corr.", respectively, we can observe that the performance variance of GPT-4 is significantly higher as the complexity of formulations increases. In 20 times of execution, GPT-4 generates the successfully executed code 18 times when implementing the medium mid_price while only three times in implementing hard alpha_pv.

| | | Avg. Exec. | Avg. Format | Avg. Corr. | Max. Corr. |
|---|---|---|---|---|---|
| GPT-4o | Data I | 0.714 | 0.330 | 0.367 | 0.540 |
| | Data II | 0.540 | 0.111 | 1.000 | 1.000 |
| | Data III | 0.778 | 0.531 | 0.422 | 0.861 |
| | Mean Value | 0.677 | 0.324 | 0.494 | 0.741 |
| LLaMa-3.1-instruct-70b | Data I | 0.690 | 0.265 | 0.239 | 0.493 |
| | Data II | 0.889 | 0.003 | 0.000 | 0.000 |
| | Data III | 0.806 | 0.569 | 0.145 | 0.261 |
| | Mean Value | 0.794 | 0.279 | 0.186 | 0.363 |
| GPT-4-turbo | Data I | 0.717 | 0.456 | 0.665 | 0.949 |
| | Data II | 0.711 | 0.056 | 0.522 | 0.556 |
| | Data III | 0.967 | 0.622 | 0.518 | 1.000 |
| | Mean Value | 0.798 | 0.378 | 0.568 | 0.835 |
| GPT-35-turbo | Data I | 0.556 | 0.100 | 0.323 | 0.453 |
| | Data II | 0.567 | 0.000 | 0.000 | 0.000 |
| | Data III | 0.767 | 0.389 | 0.431 | 0.696 |
| | Mean Value | 0.630 | 0.163 | 0.251 | 0.383 |
| Phi3-128k | Data I | 0.117 | 0.111 | 0.186 | 0.222 |
| | Data II | 0.172 | 0.000 | 0.000 | 0.000 |
| | Data III | 0.056 | 0.022 | 0.063 | 0.084 |
| | Mean Value | 0.115 | 0.044 | 0.083 | 0.102 |

Table 3: Results of large language models on RD$^2$Bench. GPT-4-turbo achieves the best performance across all metrics.

**Sources of Errors from GPT-4.** Based on the well-performed GPT-4, as an example, the first issue stems from the agents' lack of domain knowledge, which leads to erroneous operations. For instance, the model is unfamiliar with the concept of market value neutralization in the financial domain. At times, it merely defines a function without providing any content, or it directly applies a normalization operation. However, the industry-standard approach is not a simple one-step normalization but rather a multi-step data processing procedure. Through further prompting, we found that the agent knows how to do it inherently but fails to think it through during the implementation process. The second observed error occurs when the input query is lengthy. In such cases, the model often ignores specific requirements of what it should or should not do. Repeating the requirements three times usually ensures the model follows the instructions. The third observed issue is that when writing code, the model often fails to anticipate the state of the data after each step of processing. This leads to a situation where the code written for step $t + 1$ assumes the data is in the state it was at step $t - 1$, without accounting for the processing done in step $t$.

As shown in Table 3, the performance of GPT-35-turbo and Phi3-128k is poor, even failing in execution codes. However, GPT-4 models have a much better performance. This indicates that the performance of the model in the data-centric R&D task is highly related to the model's pre-training and capacity. Therefore, we posit that continually training and improving the foundation model is a promising direction for future research in the field of data-centric R&D tasks.

## 4.4 RESULTS OF MODEL ARCHITECTURE IMPLEMENTATION

In this section, we compare the performance of different LLMs in the model architecture implementation task and summarize the results in Table 4. As shown in the table, we can see the GPT-4-turbo, GPT-35-turbo-16k, and GPT-4-32K have similar running success rates but differ variously in tvc. and tsc.. The LLaMa-2-70b has the lowest running success rate and other metrics. Notice that even though a significant gap still exists between GPT-35, LLaMa-2, and GPT-4, it is much smaller than the gap in the formula implementation task. The overall running success rates are also higher than

the formula implementation task. We can conclude that we can have similar observations in the model architecture implementation task as in the formula implementation task.

| Architecture | Difficulty | Metric | GPT-4-turbo | GPT-4-32k | GPT-35-turbo-16k | LLaMa-2-70b | LLaMa-3-70b |
|---|---|---|---|---|---|---|---|
| PMLP | Easy | Avg. Exe. | 1.00 | 1.00 | 1.00 | 0.60 | 1.00 |
| | | Avg. Tsc. | **1.00** | **1.00** | 0.75 | 0.45 | 0.85 |
| | | Avg. Tvc. | **1.00** | **1.00** | 0.75 | 0.55 | 0.75 |
| | | Max. Tsc. | 1.00 | 1.00 | 1.00 | 1.00 | 1.00 |
| | | Max. Tvc. | 1.00 | 1.00 | 1.00 | 1.00 | 1.00 |
| LinkX | Easy | Avg. Exe. | **1.00** | **1.00** | **1.00** | 0.30 | **1.00** |
| | | Avg. Tsc. | **1.00** | 0.90 | 0.60 | 0.20 | 0.80 |
| | | Avg. Tvc. | 0.85 | **0.90** | 0.34 | 0.15 | 0.65 |
| | | Max. Tsc. | 1.00 | 1.00 | 1.00 | 1.00 | 1.00 |
| | | Max. Tvc. | 1.00 | 1.00 | 1.00 | 1.00 | 1.00 |
| VisNet | Hard | Avg. Exe. | **0.45** | **0.45** | 0.05 | 0.00 | 0.40 |
| | | Avg. Tsc. | **0.29** | 0.21 | 0.03 | 0.00 | 0.27 |
| | | Avg. Tvc. | 0.09 | 0.09 | 0.00 | 0.00 | **0.24** |
| | | Max. Tsc. | **0.37** | **0.37** | 0.16 | 0.00 | 0.33 |
| | | Max. Tvc. | **0.49** | **0.49** | 0.40 | 0.00 | 0.42 |
| AntiSymmetric | Medium | Avg. Exe. | **0.80** | 0.70 | 0.45 | 0.00 | **0.80** |
| | | Avg. Tsc. | **0.71** | 0.56 | 0.16 | 0.00 | 0.62 |
| | | Avg. Tvc. | 0.59 | 0.66 | 0.21 | 0.00 | **0.70** |
| | | Max. Tsc. | **0.73** | 0.66 | 0.61 | 0.00 | 0.71 |
| | | Max. Tvc. | **0.88** | **0.88** | 0.22 | 0.00 | **0.88** |
| GPSConv | Medium | Avg. Exe. | **0.75** | **0.75** | 0.45 | 0.00 | **0.75** |
| | | Avg. Tsc. | **0.56** | 0.53 | 0.24 | 0.00 | 0.51 |
| | | Avg. Tvc. | **0.62** | **0.62** | 0.19 | 0.00 | 0.59 |
| | | Max. Tsc. | **0.65** | **0.65** | 0.45 | 0.00 | **0.65** |
| | | Max. Tvc. | **1.00** | 0.72 | 0.42 | 0.00 | **1.00** |
| DirGNNConv | Medium | Avg. Exe. | **1.00** | 0.90 | 0.65 | 0.00 | 0.90 |
| | | Avg. Tsc. | **0.80** | 0.65 | 0.56 | 0.00 | 0.72 |
| | | Avg. Tvc. | **0.68** | 0.62 | 0.29 | 0.00 | **0.68** |
| | | Max. Tsc. | **0.86** | 0.82 | 0.71 | 0.00 | 0.84 |
| | | Max. Tvc. | **0.94** | 0.91 | 0.42 | 0.00 | 0.91 |

Table 4: The performance of various large language models on architecture implementation tasks.

## 5 LIMITATION

The RD$^2$Bench framework, while innovative, only evaluates the most representative base LLM, such as GPT4, GPT-4o, LLaMa-3.1, LLama3, LLama2, and GPT35, without further evaluating more open source models. Meanwhile, this paper only includesthe most representative R&D domains and problems and focuses on data-driven scenarios, which can be extended more in the future to show its generalizability. We believe that the benchmark will be a valuable tool for the community to evaluate the performance of the models in the data-centric R&D tasks and to develop new models and techniques to address the challenges and opportunities in the domain. To obtain a general cross-domain automatic R&D benchmark, we need the help of more domain experts and the participation of more research teams due to its prohibitively expensive cost.

## 6 CONCLUSION

In this paper, we serve as the first effort to tackle the real-world data-centric automatic R&D scenario in the hope of significantly improving the research efficiency of scientists and thus contributing to the revolution of human productivity. Specifically, we first propose RD$^2$Bench that benchmarks all the operations in D-CARD as a whole to navigate future work toward the ultimate goal of automating data-centric R&D directly. RD$^2$Bench focuses on evaluating the interaction and synergistic effects of various model capabilities and aiding in selecting the well-performing trustworthy models. Based on RD$^2$Bench, we find that although the most SOTA GPT-4 shows its promising potency in tackling D-CARD, there remains ample room for future work.

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

## A  FORMULA IMPLEMENTATION TASK METRICS CALCULATION DETAILS

As mentioned above, we have multiple metrics (the average and maxima score across multiple independent attempts, including "running success rate", "format success rate", "pearson correlation" and "value accuracy"). Assume the ground truth factor value is $\mathbf{Y}$ with length $n$ (the length of the time series), and the generated factor value is $\mathbf{Y}^*$, the calculation of the metrics is as follows:

**Running success** is defined as successful execution. Any error that occurs in the Python interpreter during the execution that stops the execution is considered a failure. We calculate the ratio of the number of successful execution times to the total number of attempts, denoted as avg. exe.

**Pearson correlation** is the correlation between the ground truth factor value and the generated factor value.

$$\text{corr.} = \frac{\sum_{i=1}^{n}(\mathbf{Y}_i^* - \bar{\mathbf{Y}}^*)(\mathbf{Y}_i - \bar{\mathbf{Y}})}{\sqrt{\sum_{i=1}^{n}(\mathbf{Y}_i^* - \bar{\mathbf{Y}}^*)^2}\sqrt{\sum_{i=1}^{n}(\mathbf{Y}_i - \bar{\mathbf{Y}})^2}},$$

**Format success** is defined as successful format matching, which means the final output dataframe format is (datetime, factor_name). We calculate the ratio of the number of matched result formats to the total number of attempts, denoted as avg. form.

**Value accuracy** is the accuracy of the generated factor value, which can be formulated as:

$$\text{acc.} = \frac{1}{n}\sum_{i=1}^{n}\mathbb{I}(|\mathbf{Y}_i^* - \mathbf{Y}_i| < t),$$

Please note that we set the tolerance $t$ for the value accuracy as 1e-6 in this paper, which means two values are considered as equal if the absolute difference is less than 1e-6.

## B  DATA COLLECTION DETAILS

As mentioned in the previous section, we collected papers including  (Gravina et al., 2023; Rossi et al., 2023; Rampášek et al., 2022; Lim et al., 2021; Yang et al., 2023a; Wang et al., 2024) and corresponding codes using pyg (Fey & Lenssen, 2019), which are listed in the following table.

| Paper | Type | Difficulty | GT Code |
|---|---|---|---|
| PMLP | Model | Easy | Link |
| LinkX | Model | Easy | Link |
| AntiSymmetric | Layer | Medium | Link |
| GPSConv | Layer | Medium | Link |
| DirGNNCOnv | Layer | Medium | Link |
| VisNet | Model | Hard | Link |

Table 5: Papers and corresponding ground truth implementation codes for the model architecture implementation task

## C  PROMPTS

The prompt for the model architecture implementation task is as follows:

```
The user is trying to implement some factors in quant investment,
    and you are the one to help write the Python code.
The user will provide the source data in HDF5(H5) format which you
     can load using pandas.read_hdf. The file is located near your
     Python code file which you can read from "./source_data.h5".
    After that, you will get a pandas dataframe with the following
     format:
open, close, high, low, volume, vwap, cap, IndClass.industry IndClass.
    sector, returns, date, instruments
2020-01-02,SH600000
    ,158.538132,158.538132,160.699432,158.283859,4060945.0,
159.431900,647446144.0,1.0,NaN
The explanation of the example column names:
1: returns: daily close-to-close returns
2: open, close, high, low, volume: standard definitions for daily
    price and volume data
3: vwap: daily volume-weighted average price
4: cap: market capitalization is the total value of a company's
    outstanding shares of stock
5: IndClass.industry and IndClass.sector: a generic placeholder
    for a binary industry classification such as GICS, BICS, NAICS
    , SIC, etc., in indneutralize(x, IndClass.level), where level:
     sector, industry, etc. Multiple IndClass in the same alpha
    need not correspond to the same industry classification.

The user will provide you with a formulation of the factor, which
    contains some function calls and operators. You need to
    implement the function calls and operators in Python. Your
    code is expected to align the formulation in any form which
    means the user needs to get the exact factor values with your
    code as expected.

Your code should contain the following parts: the import part, the
     function part, and the main part. You should write a main
```

function named "calculate_{function_name}" and call this
function in the "if __name__ == __main__" part. Don't write
any try-except block in your code. The user will catch the
exception message and provide feedback to you.

User will write your code into a python file and execute the file
directly with "python {your_file_name}.py". You should
calculate the factor values and save the result into an HDF5(
H5) file named "result.h5" in the same directory as your
python file. The result file is an HDF5(H5) file containing a
pandas dataframe. The index of the dataframe is the date and
instrument, and the single column name is the factor name, and
the value is the factor value. The result file should be saved
in the same directory as your python file.

To help you write the correct code, the user might provide
multiple pieces of information that help you write the correct
code:
1. The user might provide you the correct code to similar factors.
You should learn from these code to write the correct code.
2. The user might provide you the failed former code and the
corresponding feedback to the code. The feedback contains to
the execution, the code and the factor value. You should
analyze the feedback and try to correct the latest code.
3. The user might provide you with suggestions for the latest
failed code and some similar failed-to-correct pairs. Each
pair contains the failed code with a similar error and the
corresponding corrected version of the code. You should learn
from these suggestions to write the correct code.

Please respond to the code in the following JSON format. Here is
an example structure for the JSON output:
{
    "code": "The Python code as a string."
}

The prompt for the model architecture implementation task is as follows:

The user is trying to implement some models or layers in deep
learning, specifically in the graph learning area, and you are
the one to help write the Python code.

Use PyTorch and PyG (torch_geometric) framework to implement it.
You can assume the input will contain node feature X [
num_nodes, feature_dim], edge_index [2, num_edges],
edge_feature [num_edges, num_edge_features], y [num_nodes, *]
when it is node-level targets or graph-level targets of shape
[1, *], pos (node position matrix) [num_nodes, position_dim].

The user will provide you with a formulation of the model/layer.
You need to implement it in Python.

Your code should contain the following parts: the import part, the
function part, and the main part. You should write a main
function named "calculate_function_name" and call this
function in the "if __name__ == '__main__'" part. Don't write
any try-except blocks in your code. The user will catch the
exception message and provide the feedback to you.

```
User will write your code into a python file and execute the file
    directly with "python {your_file_name}.py".

Please respond with the code in the following JSON format. Here is
    an example structure for the JSON output:
{
    "code": "The Python code as a string."
}
```

## D    EXTRACTION SCHEMA

Extracted formulas and models are stored in JSON format. For each report, the schema is structured as follows:

```
{
  "report_path": "/path/to/report_1.pdf"
  "method_1": {
      "description": "the method aims to ...",  # the overall
          description of the method
      "description_figs": "/path/to/fig.png",  # the overview of
          the method, null if not found
      "formulation": ["y=\frac{1}{std}\sum...", ...],  # the
          mathematical representation of the method
      "variables": {  # the explanation of corresponding variables
          mentioned in the formulation
          "std": "the standard deviation of ...",
          "rank(x, y)": "return the largest x numbers from the
              given y numbers ..."
          ...
      },
      "parameters": {  # the value of variables given in the
          report
          "y": 16,
          ...
      },
  },
  "method_2": {},
  ...
}
```

## E    ANNOTATION DETAILS

**Annotation Tools.** In the extraction phase, annotators use a PDF editor for highlighting relevant text and VSCode for JSON editing and recording the corresponding text. In the implementation phase, annotators use Jupyter Notebook for step-by-step implementation, displaying all intermediate results. Annotated JSON and notebooks undergo the double-checking process among both annotators and senior researchers. Validated notebooks are converted into Python files and stored in the "ground_truth" folder.

**Data Management Mechanism.** Only senior researchers can move files from the "todo" and "check" folders into the "ground_truth" folder. Annotators do not have access permissions for the "ground_truth" folder.

## F    BROADER IMPACT

The proposed RD$^2$Bench has the potential to significantly impact the scientific community and industries reliant on R&D. By automating the tedious aspects of R&D, researchers can focus on

more creative and innovative aspects of their work, potentially accelerating the pace of discoveries. Smaller institutions or individual researchers with limited resources might benefit from automated tools that reduce the need for extensive human labor, making high-level R&D more accessible. Automation of R&D can reduce costs and time-to-market for new technologies, fostering faster economic growth and competitiveness

# G  MODEL PERFORMANCE ON EACH FORMULA

We exhibit the model performance on each formula in the following tables.

| Data | Difficulty | Formula | Avg. Exec. | Avg. Format | Avg. Corr. | Max. Corr. |
|---|---|---|---|---|---|---|
| Data I | Easy | PB_ROE | 0.650 | 0.050 | 0.852 | 0.852 |
| | | PB_ROE_2 | 0.600 | 0.200 | 0.875 | 1.000 |
| | | PB_ROE_3 | 0.600 | 0.300 | 0.726 | 1.000 |
| | Medium | ROE_movement | 0.950 | 0.750 | 0.934 | 1.000 |
| | | ROE_movement_10 | 0.900 | 0.800 | 0.803 | 1.000 |
| | | ROE_movement_20 | 0.950 | 0.750 | 0.703 | 1.000 |
| | Hard | PB_ROE_movement | 0.600 | 0.450 | 0.516 | 0.897 |
| | | PB_ROE_movement_10 | 0.650 | 0.300 | 0.327 | 0.896 |
| | | PB_ROE_movement_20 | 0.550 | 0.500 | 0.244 | 0.896 |
| Data II | Easy | mid_price | 0.800 | 0.100 | 1.000 | 1.000 |
| | | mid_price_2 | 0.850 | 0.000 | NaN | NaN |
| | | mid_price_3 | 0.850 | 0.000 | NaN | NaN |
| | Medium | liquidity_imbalance | 0.500 | 0.050 | 1.000 | 1.000 |
| | | liquidity_imbalance_2 | 0.900 | 0.150 | 0.694 | 1.000 |
| | | liquidity_imbalance_3 | 0.450 | 0.100 | 1.000 | 1.000 |
| | Hard | micro_price | 0.850 | 0.000 | NaN | NaN |
| | | micro_price_2 | 0.600 | 0.000 | NaN | NaN |
| | | micro_price_3 | 0.600 | 0.100 | 1.000 | 1.000 |
| Data III | Easy | alpha053 | 0.950 | 0.700 | 0.933 | 1.000 |
| | | alpha053_15 | 0.950 | 0.650 | 0.872 | 1.000 |
| | | alpha053_5 | 1.000 | 0.650 | 0.676 | 1.000 |
| | Medium | alpha_pv_diff | 1.000 | 0.600 | 0.513 | 1.000 |
| | | alpha_pv_diff_15 | 0.950 | 0.750 | 0.258 | 1.000 |
| | | alpha_pv_diff_20 | 1.000 | 0.750 | 0.441 | 1.000 |
| | Hard | alpha_pv_diff_pct | 0.950 | 0.700 | 0.375 | 1.000 |
| | | alpha_pv_diff_pct_15 | 0.900 | 0.450 | 0.236 | 1.000 |
| | | alpha_pv_diff_pct_20 | 1.000 | 0.350 | 0.358 | 1.000 |
| Overall | N/A | Avg. Data I | 0.717 | 0.456 | 0.665 | 0.949 |
| | | Avg. Data II | 0.711 | 0.056 | 0.522 | 0.556 |
| | | Avg. Data III | 0.967 | 0.622 | 0.518 | 1.000 |
| | | Mean Value | 0.798 | 0.378 | 0.568 | 0.835 |

Table 6: The performance of GPT-4-turbo in formula implementation.

| Category | Difficulty | Factor | avg. exec. | avg. format | avg. corr. | max. corr. |
|---|---|---|---|---|---|---|
| Fundamentals | Easy | PB_ROE | 0.643 | 0.143 | 0.844 | 0.844 |
| | | PB_ROE_2 | 0.571 | 0.000 | NaN | NaN |
| | | PB_ROE_3 | 0.571 | 0.222 | 0.382 | 0.764 |
| | Hard | PB_ROE_movement | 0.714 | 0.375 | 0.036 | 0.039 |
| | | PB_ROE_movement_10 | 0.714 | 0.375 | 0.233 | 0.619 |
| | | PB_ROE_movement_20 | 0.929 | 0.659 | 0.012 | 0.039 |
| | Medium | ROE_movement | 0.786 | 0.500 | 0.016 | 0.016 |
| | | ROE_movement_10 | 0.714 | 0.500 | 0.412 | 1.000 |
| | | ROE_movement_20 | 0.786 | 0.200 | 1.000 | 1.000 |
| High-Frequency | Easy | mid_price | 0.571 | 0.250 | 1.000 | 1.000 |
| | | mid_price_2 | 0.286 | 0.042 | NaN | NaN |
| | | mid_price_3 | 0.643 | 0.000 | NaN | NaN |
| | Hard | micro_price | 0.500 | 0.111 | 1.000 | 1.000 |
| | | micro_price_2 | 0.643 | 0.125 | NaN | NaN |
| | | micro_price_3 | 0.500 | 0.055 | NaN | NaN |
| | Medium | liquidity_imbalance | 0.714 | 0.143 | NaN | NaN |
| | | liquidity_imbalance_2 | 0.429 | 0.050 | NaN | NaN |
| | | liquidity_imbalance_3 | 0.571 | 0.222 | 1.000 | 1.000 |
| Volume&Price | Easy | alpha053 | 0.643 | 0.000 | NaN | NaN |
| | | alpha053_15 | 0.571 | 0.000 | NaN | NaN |
| | | alpha053_5 | 0.143 | 0.071 | 1.000 | 1.000 |
| | Hard | alpha_pv_diff_pct | 0.786 | 0.553 | 0.500 | 1.000 |
| | | alpha_pv_diff_pct_15 | 1.000 | 0.880 | 0.201 | 1.000 |
| | | alpha_pv_diff_pct_20 | 0.929 | 0.790 | 0.501 | 1.000 |
| | Medium | alpha_pv_diff | 1.000 | 0.825 | 0.025 | 0.025 |
| | | alpha_pv_diff_15 | 0.929 | 0.778 | 0.294 | 1.000 |
| | | alpha_pv_diff_20 | 1.000 | 0.884 | 0.433 | 1.000 |

Table 7: The performance of gpt-4o in formula implementation.

| Category | Difficulty | Factor | avg. exec. | avg. format | avg. corr. | max. corr. |
|---|---|---|---|---|---|---|
| Fundamentals | Easy | PB_ROE | 0.643 | 0.071 | NaN | NaN |
| | | PB_ROE_2 | 0.643 | 0.143 | 0.182 | 0.182 |
| | | PB_ROE_3 | 0.429 | 0.000 | NaN | NaN |
| | Hard | PB_ROE_movement | 0.571 | 0.125 | 0.668 | 0.668 |
| | | PB_ROE_movement_10 | 0.571 | 0.125 | 0.295 | 0.295 |
| | | PB_ROE_movement_20 | 0.714 | 0.167 | 0.009 | 0.009 |
| | Medium | ROE_movement | 0.929 | 0.750 | 0.181 | 1.000 |
| | | ROE_movement_10 | 0.857 | 0.600 | 0.186 | 0.999 |
| | | ROE_movement_20 | 0.857 | 0.400 | 0.151 | 0.298 |
| High-Frequency | Easy | mid_price | 0.929 | 0.002 | NaN | NaN |
| | | mid_price_2 | 0.786 | 0.004 | NaN | NaN |
| | | mid_price_3 | 0.929 | 0.001 | NaN | NaN |
| | Hard | micro_price | 0.857 | 0.003 | NaN | NaN |
| | | micro_price_2 | 0.857 | 0.004 | NaN | NaN |
| | | micro_price_3 | 1.000 | 0.002 | NaN | NaN |
| | Medium | liquidity_imbalance | 0.929 | 0.001 | NaN | NaN |
| | | liquidity_imbalance_2 | 0.857 | 0.003 | NaN | NaN |
| | | liquidity_imbalance_3 | 0.857 | 0.003 | NaN | NaN |
| Volume&Price | Easy | alpha053 | 0.929 | 0.667 | 0.455 | 1.000 |
| | | alpha053_15 | 1.000 | 0.857 | 0.145 | 0.301 |
| | | alpha053_5 | 0.357 | 0.286 | 0.659 | 1.000 |
| | Hard | alpha_pv_diff_pct | 0.857 | 0.667 | 0.001 | 0.001 |
| | | alpha_pv_diff_pct_15 | 0.786 | 0.556 | 0.000 | 0.001 |
| | | alpha_pv_diff_pct_20 | 0.857 | 0.700 | 0.003 | 0.004 |
| | Medium | alpha_pv_diff | 0.857 | 0.500 | 0.025 | 0.025 |
| | | alpha_pv_diff_15 | 0.929 | 0.613 | 0.011 | 0.011 |
| | | alpha_pv_diff_20 | 0.643 | 0.278 | 0.008 | 0.008 |

Table 8: The performance of LLaMa-3.1-70B-Instruct in formula implementation.

|  |  |  | avg. exec. | avg. format | avg. corr. | max. corr. |
|---|---|---|---|---|---|---|
| Fundamental | Easy | PB_ROE | 0.400 | 0.000 | NaN | NaN |
|  |  | PB_ROE_2 | 0.600 | 0.000 | NaN | NaN |
|  |  | PB_ROE_3 | 0.600 | 0.200 | 0.521 | 0.999 |
|  | Medium | ROE_movement | 0.800 | 0.300 | 0.339 | 1.000 |
|  |  | ROE_movement_10 | 0.600 | 0.100 | 1.000 | 1.000 |
|  |  | ROE_movement_20 | 0.900 | 0.200 | 0.967 | 1.000 |
|  | Hard | PB_ROE_movement | 0.200 | 0.100 | 0.078 | 0.078 |
|  |  | PB_ROE_movement_10 | 0.500 | 0.000 | NaN | NaN |
|  |  | PB_ROE_movement_20 | 0.400 | 0.000 | NaN | NaN |
| High Frequency | Easy | mid_price | 0.600 | 0.000 | NaN | NaN |
|  |  | mid_price_2 | 0.500 | 0.000 | NaN | NaN |
|  |  | mid_price_3 | 0.600 | 0.000 | NaN | NaN |
|  | Medium | liquidity_imbalance | 0.200 | 0.000 | NaN | NaN |
|  |  | liquidity_imbalance_2 | 0.800 | 0.000 | NaN | NaN |
|  |  | liquidity_imbalance_3 | 0.500 | 0.000 | NaN | NaN |
|  | Hard | micro_price | 0.400 | 0.000 | NaN | NaN |
|  |  | micro_price_2 | 0.700 | 0.000 | NaN | NaN |
|  |  | micro_price_3 | 0.800 | 0.000 | NaN | NaN |
| Price Volume | Easy | alpha053 | 0.800 | 0.500 | 0.809 | 1.000 |
|  |  | alpha053_15 | 0.700 | 0.500 | 0.806 | 1.000 |
|  |  | alpha053_5 | 0.700 | 0.500 | 0.440 | 1.000 |
|  | Medium | alpha_pv_diff | 0.800 | 0.700 | 0.304 | 1.000 |
|  |  | alpha_pv_diff_15 | 0.700 | 0.400 | 0.259 | 1.000 |
|  |  | alpha_pv_diff_20 | 0.600 | 0.400 | 1.000 | 1.000 |
|  | Hard | alpha_pv_diff_pct | 0.800 | 0.200 | -0.011 | -0.011 |
|  |  | alpha_pv_diff_pct_15 | 0.900 | 0.200 | 0.096 | 0.096 |
|  |  | alpha_pv_diff_pct_20 | 0.900 | 0.100 | 0.176 | 0.176 |
| gpt3.5 | N/A | Fundamental Avg | 0.556 | 0.100 | 0.323 | 0.453 |
|  |  | High Frequency Avg | 0.567 | 0.000 | 0.000 | 0.000 |
|  |  | Price Volume Avg | 0.767 | 0.389 | 0.431 | 0.696 |
|  |  | mean value (0 for NaN) | 0.630 | 0.163 | 0.251 | 0.383 |

Table 9: The performance of gpt-35-turbo in formula implementation.

| | | | avg. exec. | avg. format | avg. corr. | max. corr. |
|---|---|---|---|---|---|---|
| Fundamental | Easy | PB_ROE | 0.000 | 0.000 | NaN | NaN |
| | | PB_ROE_2 | 0.050 | 0.000 | NaN | NaN |
| | | PB_ROE_3 | 0.000 | 0.000 | NaN | NaN |
| | Medium | ROE_movement | 0.350 | 0.350 | 1.000 | 1.000 |
| | | ROE_movement_10 | 0.350 | 0.350 | 0.675 | 1.000 |
| | | ROE_movement_20 | 0.300 | 0.300 | NaN | NaN |
| | oo Hard | PB_ROE_movement | 0.000 | 0.000 | NaN | NaN |
| | | PB_ROE_movement_10 | 0.000 | 0.000 | NaN | NaN |
| | | PB_ROE_movement_20 | 0.000 | 0.000 | NaN | NaN |
| High Frequency | Easy | mid_price | 0.250 | 0.000 | NaN | NaN |
| | | mid_price_2 | 0.250 | 0.000 | NaN | NaN |
| | | mid_price_3 | 0.400 | 0.000 | NaN | NaN |
| | Medium | liquidity_imbalance | 0.050 | 0.000 | NaN | NaN |
| | | liquidity_imbalance_2 | 0.150 | 0.000 | NaN | NaN |
| | | liquidity_imbalance_3 | 0.450 | 0.000 | NaN | NaN |
| | Hard | micro_price | 0.000 | 0.000 | NaN | NaN |
| | | micro_price_2 | 0.000 | 0.000 | NaN | NaN |
| | | micro_price_3 | 0.000 | 0.000 | NaN | NaN |
| Price Volume | Easy | alpha053 | 0.050 | 0.000 | NaN | NaN |
| | | alpha053_15 | 0.000 | 0.000 | NaN | NaN |
| | | alpha053_5 | 0.050 | 0.000 | NaN | NaN |
| | Medium | alpha_pv_diff | 0.250 | 0.150 | 0.413 | 0.602 |
| | | alpha_pv_diff_15 | 0.050 | 0.000 | NaN | NaN |
| | | alpha_pv_diff_20 | 0.000 | 0.000 | NaN | NaN |
| | Hard | alpha_pv_diff_pct | 0.050 | 0.050 | 0.153 | 0.153 |
| | | alpha_pv_diff_pct_15 | 0.000 | 0.000 | NaN | NaN |
| | | alpha_pv_diff_pct_20 | 0.050 | 0.000 | NaN | NaN |
| phi3_128k | N/A | Fundamental Avg | 0.117 | 0.111 | 0.186 | 0.222 |
| | | High Frequency Avg | 0.172 | 0.000 | 0.000 | 0.000 |
| | | Price Volume Avg | 0.056 | 0.022 | 0.063 | 0.084 |
| | | mean value (0 for NaN) | 0.115 | 0.044 | 0.083 | 0.102 |

Table 10: The performance of Phi3-128k in formula implementation.

