# OpenReview forum: "RD2Bench: Toward Data-Centric Automatic R&D"
_ICLR.cc/2025/Conference — Submitted to ICLR 2025_

### Official Review · Reviewer_JWbd · 2024-10-22

**Soundness:** 3
**Presentation:** 3
**Contribution:** 3
**Rating:** 5
**Confidence:** 3

**Summary:**

This paper introduces RD2Bench, a benchmark designed to enhance real-world data-centric automatic R&D (D-CARD) by evaluating the synergistic effects of various model capabilities. The authors aim to improve research efficiency and foster the automation of data-centric R&D processes. The study highlights the strengths of state-of-the-art models like GPT-4 while identifying areas for further improvement.

**Strengths:**

1. Innovative approach to automating data-centric R&D with a comprehensive benchmark.
2. Thorough methodology encompassing data collection, human annotation, and robust evaluation metrics.
3. Insightful experimental results demonstrating the capabilities and limitations of current models.

**Weaknesses:**

1. Data is limited to the financial domain; a broader variety of datasets would enhance the benchmark's applicability.
2. The paper would benefit from providing related implementation codes and datasets to facilitate reproducibility.
3. There is a lack of comparison with model combinations, which could provide insights into potential performance improvements.
4. Insufficient discussion on the limitations and potential biases in the data collection process.
5. Need for more clarity on how the benchmark could be adapted for various domains beyond the current focus.

**Questions:**

See above.

---

### Official Review · Reviewer_CkQf · 2024-10-26

**Soundness:** 2
**Presentation:** 3
**Contribution:** 2
**Rating:** 5
**Confidence:** 3

**Summary:**

This paper introduces a benchmark called "RD2Bench" designed to evaluate the process of Data-Centric Automatic R&D (D-CARD). RD2Bench focuses on the interaction and synergy between the abilities of large language models (LLMs) in areas such as language understanding, code implementation, and data selection during the R&D process. The benchmark simulates real-world data-centric R&D scenarios to study the performance and challenges of these models in handling automated R&D tasks. Experimental results indicate that while current state-of-the-art LLMs perform well on some simple R&D tasks, more complex tasks still pose significant challenges.

**Strengths:**

1. The paper is the first to formalize the task of automating research and development (R&D), which is an important and impactful endeavor.
2. The motivation of the paper is well-grounded and clearly articulated.

**Weaknesses:**

1. The dataset size and scope are quite limited, containing only 27 formulas and 6 models. The formulas are solely from the financial domain, and the models are only graph neural networks, representing a very small part of "Data-Centric Research and Development.
2. The authors claim RD2Bench evaluates the interaction and synergy of various model capabilities, but the results only assess "information extraction" and "method implementation" separately. Existing benchmarks, like [1], already combine these capabilities, and RD2Bench's unique contribution in this regard is not clearly demonstrated.
3. The evaluation metrics, task descriptions, and several experimental details remain unclear, particularly with the absence of Appendix A.

[1] Mialon, G., Fourrier, C., Swift, C., Wolf, T., LeCun, Y., & Scialom, T. (2023). Gaia: a benchmark for general ai assistants. arXiv preprint arXiv:2311.12983.

**Questions:**

1.	Is there a more detailed explanation of the task difficulty levels?
2.	Regarding method extraction in Section 3.3: Figures 1 and 2 only show the formula extraction results. Are the model extraction results also in a key-value format? What exactly do the evaluation metrics measure? Is it the matching degree between expected keys and extracted keys?
3.	In the Implementation Step, are the method extraction ground truths used as inputs, or are the actual extracted results (even if potentially incorrect) used as inputs?

---

### Official Review · Reviewer_4AzY · 2024-11-04

**Soundness:** 4
**Presentation:** 3
**Contribution:** 4
**Rating:** 8
**Confidence:** 4

**Summary:**

This paper introduces RD2Bench, a benchmark designed to evaluate and advance data-driven automated R&D processes. RD2Bench assesses large language models on their performance in automated R&D tasks, focusing on two core abilities: (1) language understanding to accurately extract implementable methods from research materials, and (2) technical implementation skills to develop reliable and trustworthy solutions. Unlike existing benchmarks, RD2Bench evaluates not just individual capabilities but also the synergistic effects of multiple abilities in real-world R&D tasks. The contributions of the paper include providing a structured framework to assess model performance in automated R&D workflows and establishing metrics to measure task success, accuracy, and stability. Experimental results reveal the potential of advanced models like GPT-4 in these tasks, while also identifying areas needing further improvement to achieve full automation in scientific and data-driven fields.

**Strengths:**

1. Innovative and Practical Benchmark: RD2Bench introduces a unique data-centric benchmark specifically designed for automating R&D tasks, moving beyond traditional evaluations by assessing multiple model capabilities in real-world applications.

2. Focus on Synergistic Model Abilities: Unlike other benchmarks that assess isolated abilities, RD2Bench emphasizes the interaction and synergy between language understanding and technical implementation, providing a comprehensive evaluation framework for large language models.

3. Detailed and Insightful Experimental Findings: The paper provides experimental results that reveal specific strengths and weaknesses of state-of-the-art models like GPT-4, offering actionable insights for future research on improving automated R&D capabilities.

4. Robust Metric Design: The benchmark incorporates multiple detailed metrics, such as running success rate, format success rate, correlation, and accuracy, allowing nuanced assessments of model performance across diverse R&D tasks.

5. Structured Literature Review: The paper contextualizes its contributions within existing research on language models and automated R&D, effectively highlighting gaps and motivating the need for a new benchmark like RD2Bench.

6. Practical Applications for Productivity Enhancement: RD2Bench has clear practical implications, as it is designed to improve the efficiency and productivity of scientific research processes, making it highly relevant for both academic and industrial applications.

**Weaknesses:**

1. Lack of Analysis on Model Error Sources: While performance metrics like accuracy and correlation are well-covered, the paper could strengthen its impact by analyzing common sources of model errors, such as misunderstanding prompts or misinterpreting data. A deeper error analysis could highlight specific improvement areas in LLMs for R&D.

2. Limited Detail on Human Annotation Quality Control: Although human annotation plays a significant role in RD2Bench, the paper could provide more information on quality control mechanisms for annotators beyond training and double-checking. Expanding on inter-annotator agreement or including validation checks would reinforce the reliability of the benchmark.

3.Insufficient Discussion on Computational Efficiency: RD2Bench implies the need for repeated trials and extensive processing, especially for large models. An analysis of the benchmark’s computational requirements and possible optimizations could make it more practical for widespread research use, especially in settings with limited resources.

**Questions:**

1. How do the authors envision RD2Bench being adapted or extended for domain-specific applications, such as finance or biotech, where specialized knowledge is often critical? Are there plans to incorporate domain-specific adaptations in future work?

2. Could the authors elaborate on the complexity of the R&D tasks included in RD2Bench? Introducing more advanced, iterative R&D tasks could better reflect real-world challenges. Would the authors consider integrating more complex task workflows?

3. What are the most common sources of errors encountered in the benchmark tasks, and how could they inform improvements for language models? An in-depth error analysis might reveal valuable insights into model limitations and areas for enhancement.

4. The paper mentions quality control for human annotations but provides limited details. Could the authors clarify the mechanisms used to ensure annotation consistency, such as inter-annotator agreement metrics or other quality checks?

5. Given the computational demands of RD2Bench, have the authors considered optimizations to reduce resource requirements? An analysis of potential optimizations would be beneficial for research environments with constrained computational resources.

---

### Official Review · Reviewer_HBxj · 2024-11-04

**Soundness:** 2
**Presentation:** 3
**Contribution:** 2
**Rating:** 3
**Confidence:** 5

**Summary:**

The paper proposes a benchmark named RD2Bench for evaluating data-centric automatic Research & Development (R&D) using Large Language Models (LLMs). The benchmark aims to assess and improve the efficiency of the R&D process by evaluating a model’s ability to understand, extract, and implement methods described in raw research materials. The focus is on leveraging LLMs to perform data-centric R&D in a real-world context, with an emphasis on minimizing manual labor for researchers. The authors also present an evaluation of current state-of-the-art LLMs, like GPT-4, in different stages of data-centric R&D.

**Strengths:**

1. This paper propose an interesting task. The concept of automating the R&D process to minimize human intervention is innovative and highly valuable.
2. The proposed RD2Bench appears well-conceived, focusing not only on a model’s understanding ability but also on the complex interaction between multiple abilities, such as data extraction, method selection, and implementation.

**Weaknesses:**

This paper presents an interesting and novel task for large language models, contributing valuable insights to the field. However, despite its strengths, there are still some weaknesses that prevent me from assigning a higher score and lead me to believe that the work, in its current form, is not yet sufficiently developed or complete.
1. While the goal of RD2Bench is to evaluate models across a broad spectrum of R&D tasks, the current focus on only financial reports and stock trading data is a significant limitation. The models' performance on financial data may not be indicative of how well they would perform in fields with different data characteristics or domain-specific challenges.
2. The paper mentions that RD2Bench includes human-annotated ground-truth results, but it provides insufficient detail about the annotation guidelines and mechanisms used. Given that human annotation quality is vital for evaluating automated systems, a more comprehensive explanation of how annotation challenges were overcome would be valuable.
3. The performance metrics reported in Tables 1, 2, and 4 show values that are frequently close to 0.9 or even nearly 1.0, which raises concerns about the benchmark's effectiveness in distinguishing the capabilities of different models. Such high scores across various models suggest that the benchmark may lack the complexity or sensitivity needed to reveal meaningful performance differences, potentially limiting its utility for assessing model strengths and weaknesses comprehensively.

**Questions:**

1. The paper evaluates the performance of GPT-4 and Llama but does not extend the analysis to other state-of-the-art models such as GPT-4o, Claude, or a broader selection of open-source alternatives. Including these models could provide a more comprehensive assessment of current capabilities in automatic R&D.
2. The study's focus on financial domain data raises concerns about the generalizability of the benchmark. Without empirical evaluation across diverse domains, it remains unclear whether the models would exhibit comparable performance in other research areas, which limits the applicability of the findings.

---

> ### Comment · Reviewer_HBxj · 2024-11-30
>
> Thank you for your feedback and the additional results! And thank you for adding the annotation details. However, I still have several concerns. First, I encourage the authors to thoroughly review all the state-of-the-art models covered in other related benchmark papers. While I have mentioned two SOTA models, testing only five models is insufficient for a comprehensive benchmark study. Additionally, although the authors have revised Section 3, after reviewing the introduction and abstract, I remain concerned about the potential for overselling the work. I would strongly encourage the authors to extend the scope of the paper to include more than two domains.
>
> I believe this research problem is worth further exploration. However, my current concerns cannot be addressed within a short time frame. I hope this work can become more complete in the future. As such, I will maintain my current score.

---

> ### Comment · Reviewer_HBxj · 2024-12-02
>
> Thank you for your response. I suggest examining other highly-rated benchmarks from the ICLR2025 submissions for additional context. For example:
>
> + Submission7553 (BigCodeBench): Benchmarks 60 LLMs.
> + Submission5074 (Cybench): Similar to your approach, it evaluates agentic workflows on 8 models for complex tasks.
> + Submission264 (LOKI): Benchmarks 22 open-source LMMs and 6 closed-source models.
>
> These benchmarks assess complex tasks on a diverse range of leading models. On a related note, I’m unclear why, given the availability of GPT-4o, you still refer to GPT-4-turbo and GPT-3.5-turbo as SOTA LLMs. Without these two older versions, your evaluation includes **only 3 SOTA LLMs**. Furthermore, I question whether Phi3-128k qualifies as a "SOTA LLM," as it is rarely included in other recently-released benchmarks.
>
> Regarding the second point, I previously suggested limiting your paper’s scope to the financial domain. While the revised abstract partially addresses the overselling issue, I still believe the title "RD2Bench" is too ambitious for a benchmark focused solely on the financial domain, as it does not reflect the broader scope implied by the name.
>
> I don't think the authors' responses have adequately addressed my concerns for the reasons outlined above. Emphasizing that their work has garnered attention or is resource-intensive does not, in itself, justify it being considered of high quality or deserving of acceptance.
>
> With a responsible attitude, I believe this work has proposed a very interesting direction that is worth exploring. However, I think the current work, as a benchmark, lacks exploration of more models and domains and needs further improvement. Therefore, I still hold a reject stance.

---

> ### Comment · Reviewer_HBxj · 2024-12-03
>
> Thank you for the reply.
>
> Have you thoroughly reviewed the paper I referenced? How can you dismiss **Claude, Gemini, Mistral, Deepseek, Qwen, InternLM, and numerous other state-of-the-art (SOTA) models**, claiming they are not SOTA enough to "provide useful information"? Additionally, do you genuinely believe that testing on just three SOTA models is sufficient for a benchmark paper?
>
> Shouldn't you take greater responsibility for both your paper and its audience? As a benchmark study, the only notable aspect of your work is the interesting task and pipeline you proposed. Beyond that, **the paper is far from meeting the standards expected at the present time.**

---

> ### Comment · Reviewer_HBxj · 2024-12-03
>
> Thank you for your response. Please note that we are now at the end of 2024. I am unclear why you continue to reference benchmark papers for 2023 submitting conferences. The field of large language models is evolving rapidly, yet your benchmarks are still based on non-SOTA models. Despite claiming to test all meaningful SOTA models, you disregard my suggestions to include models such as Claude, Gemini, Mistral, DeepSeek, Qwen, InternLM, and numerous other cutting-edge models.
>
> I find it puzzling that you believe your work,with its limited experimental results confined to narrow domains and testing on only three SOTA LLMs, has the potential to "inspire others, provide reliable information, and advance the field and subsequent research." All my feedback is grounded in the expectations for benchmark papers submitted to top-tier computer science conferences near 2025. If you think a paper with inspiring ideas but insufficient experiments and results can qualify as a benchmark submission, I strongly encourage you to consider conferences that are not top-tier. Compared to other **ICLR 2025 benchmark submission papers**, I don’t believe your current work meets the standards expected of a top-tier conference.

---

> ### Comment · Reviewer_HBxj · 2024-12-03
>
> Thank you for your response. I have carefully reviewed the papers you mentioned. Here are my observations:
>
> + DevBench: This is not an LLM benchmark. It evaluates the capabilities of a diverse set of vision-language models, which falls outside the scope of your study.
>
> + LINGOLY: This benchmark evaluates 12 large language models, including Llama 3 (8B and 70B), Mixtral 8x7B, Aya 23 35B, Gemma 7B, Llama 2 70B, GPT-4o, GPT-4, GPT-3.5, Claude Opus, and Gemini 1.5 Pro. This encompasses most of the models I mentioned, far exceeding the number of models you tested.
>
> + CVQA: This benchmark assesses the capabilities of multimodal language models (MLLM), which differs from your focus on pure text-based LLMs. Moreover, it includes tests on Gemini, a model I pointed out but which is absent in your work.
>
> + MedCalc-Bench: This is a domain-specific benchmark that diverges slightly from general benchmarks. It evaluates medical domain-specific LLMs such as PMC-LLaMA-13B and MEDITRON-70B, alongside proprietary LLMs like GPT-4 and GPT-3.5, and open-source LLMs including Llama 3 (8B and 70B), Mistral-7B, and Mixtral-8x7B. Again, the number and diversity of models tested significantly exceed what you evaluated.
>
> + ChaosBench: This is not an LLM benchmark but instead focuses on cutting-edge ML-based weather emulators, which is entirely outside your study's scope.
>
> **The ICLR 2025 submission papers I mentioned align more closely with the scope of your work.** Beyond the differences in the range of tested models, your benchmarks also fall short in terms of scale and content diversity compared to those in these papers you listed above. Your work falls short of the standards expected at top-tier conferences. **Could you please stop justifying your work with unrelated studies and instead consider how to make your work more comprehensive?**
>
> I initially appreciated the idea you presented. While I found your work insufficient, I believed the direction had potential and was worth pursuing, which is why I provided numerous advice to help improve it. However, the actual work you have delivered, along with your rebuttal responses, has been deeply disappointing. It appears that instead of focusing on refining and strengthening your work to make it more comprehensive, you have prioritized citing unrelated papers to mislead reviewers and the audience in an attempt to secure acceptance.
>
> **For these reasons, and after reviewing additional related papers, I will increase my confidence level to 5.** I strongly believe your work does not meet the standards required for submission to top-tier conferences. I encourage you to reflect on how to refine your paper in the future. **Gaining acceptance is far less meaningful than producing work that is truly insightful and impactful.**

---

> ### Comment · Reviewer_HBxj · 2024-12-03
>
> Thank you for your response.
>
> + Regarding your first response: I specifically emphasized the issue of unrelated studies when I stated, "stop justifying your work with **unrelated studies.**" My concern lies with the irrelevance of the cited studies. It is undeniable that substantial evidence for LLM benchmarks exists, published in top-tier conferences such as LINGOLY, which **have evaluated significantly more LLMs compared to your work**, as previously outlined. In my initial review, I suggested "extending the analysis to other state-of-the-art models, such as GPT-4o, Claude, or a broader selection of open-source alternatives." However, there are still no experimental results provided for **even Claude**. While GPT-4o and Claude were examples, my broader concern is the insufficiency of the current evaluation for a benchmark paper. My critique was not a request for additional experiments but rather an observation that the current experimental scope is far too limited.
>
> + Regarding your second response: As noted earlier, all the benchmarks you referenced—even those **outside your specific study scope**—evaluated a sufficient number of models within their respective contexts (e.g., ML models, MLLMs, VLMs, domain-specific LLMs). In contrast, your paper evaluates only three state-of-the-art LLMs. The claim that "scaling the number of evaluated models is not correlated with the quality of a benchmark paper" seems unsubstantiated, particularly when **the examples provided show the opposite—comprehensive experiments across numerous models.**
>
> + Regarding your third and fourth responses: Your justification does not effectively address the concerns raised. Listing numerous unrelated papers to **obscure the issue** does not enhance your work and is not a constructive approach to engaging with reviewers. **As a responsible reviewer, I carefully evaluated your responses and provided detailed suggestions for improving your paper. I also reviewed the works you cited.** However, your responses focus largely on asserting that your work is adequate for a top-tier conference, supported by unrelated studies, rather than incorporating the suggestions I provided. All the suggestions I have provided are based on my observations of currently released LLM benchmarks, which I believe are worth referencing. This does not reflect a genuine effort to address the feedback meaningfully, despite your claim that 'you have tried your best to reach this goal in the whole conversation process.'
>
> Additionally, I noticed that your responses are not visible to the audience, only to reviewers. I am unsure if this aligns with the principles of OpenReview.

---

### Meta-Review · Area_Chair_CxBF · 2024-12-19

**Metareview:**

This paper introduces RD2Bench, which is a benchmark designed to advance data-driven automated R&D processes. RD2Bench proposed the method of extracting and implementing models from raw information like financial reports and ML papers and evaluates the ability of several language models (LLMs).

The paper has received mixed scores: one positive (8) and three negatives (5, 5, 3). Based on several rounds of discussions between reviewers, there are still some issues in areas such as model evaluation and Implementation details. Therefore, the final decision is to reject the paper. We hope the authors will revise it in the subsequent version according to the reviewers' comments.

The specific reasons are as follows:
1. It is necessary to evaluate more models to demonstrate the validity of this work. There is still room for improvement in the model testing section of the paper.
2. The paper has numerous details that need to be improved and supplemented, such as dataset annotations, cost, and optimizations regarding computational requirements.

**Additional Comments On Reviewer Discussion:**

The paper has received mixed scores: one positive (8) and three negatives (5, 5, 3). The discussion between the authors and reviewers regarding the main concerns is as follows:

Reviewers HBxj, 4AzY, CkQf, and JWbd all raised concerns about the scope of data application in the paper (currently, the data is limited to the financial domain).  In response, the authors argue that data from the financial domain is more representative, with clearer evaluation methods and transferability.

Reviewers HBxj and CkQf mentioned the limitation in the number of models and engaged in discussion with the authors.  Especially reviewer HBxj emphasized the inadequate evaluation of large language models in this paper, where comprehensive evaluations across a diverse range of models are required for a benchmark paper.

---

### Decision · Program_Chairs · 2025-01-22

Reject